# MLK3 Regulates Inflammatory Response via Activation of AP-1 Pathway in HEK293 and RAW264.7 Cells

**DOI:** 10.3390/ijms231810874

**Published:** 2022-09-17

**Authors:** Anh Thu Ha, Jae Youl Cho, Daewon Kim

**Affiliations:** 1Department of Integrative Biotechnology, Biomedical Institute for Convergence at SKKU (BICS), Sungkyunkwan University, Suwon 16419, Korea; 2Laboratory of Bio-Informatics, Department of Multimedia Engineering, Dankook University, Yongin 16890, Korea

**Keywords:** MLK3, inflammation, AP-1 pathway, macrophages, MAPK

## Abstract

Inflammation is a critically important barrier found in innate immunity. However, severe and sustained inflammatory conditions are regarded as causes of many different serious diseases, such as cancer, atherosclerosis, and diabetes. Although numerous studies have addressed how inflammatory responses proceed and what kinds of proteins and cells are involved, the exact mechanism and protein components regulating inflammatory reactions are not fully understood. In this paper, to determine the regulatory role of mixed lineage kinase 3 (MLK3), which functions as mitogen-activated protein kinase kinase kinase (MAP3K) in cancer cells in inflammatory response to macrophages, we employed an overexpression strategy with MLK3 in HEK293 cells and used its inhibitor URMC-099 in lipopolysaccharide (LPS)-treated RAW264.7 cells. It was found that overexpressed MLK3 increased the mRNA expression of inflammatory genes (COX-2, IL-6, and TNF-α) via the activation of AP-1, according to a luciferase assay carried out with AP-1-Luc. Overexpression of MLK3 also induced phosphorylation of MAPKK (MEK1/2, MKK3/6, and MKK4/7), MAPK (ERK, p38, and JNK), and AP-1 subunits (c-Jun, c-Fos, and FRA-1). Phosphorylation of MLK3 was also observed in RAW264.7 cells stimulated by LPS, Pam3CSK, and poly(I:C). Finally, inhibition of MLK3 by URMC-099 reduced the expression of COX-2 and CCL-12, phosphorylation of c-Jun, luciferase activity mediated by AP-1, and phosphorylation of MAPK in LPS-treated RAW264.7 cells. Taken together, our findings strongly suggest that MLK3 plays a central role in controlling AP-1-mediated inflammatory responses in macrophages and that this enzyme can serve as a target molecule for treating AP-1-mediated inflammatory diseases.

## 1. Introduction

Inflammation is a self-defense mechanism that protects the body against external attacks by bacteria, fungi, viruses, and protozoa. In addition, when the body is physically damaged, its inflammatory reactions can be activated to restore the damage [1,2,3]. Thus, without inflammatory responses, the body is unable to maintain healthy homeostatic conditions.

Due to extensive studies, how external and intrinsic damage signals can activate the body’s inflammation is understood. To recognize dangerous signaling, inflammatory cells such as macrophages and neutrophils express pattern recognition receptors (PRRs) such as toll-like receptors (TLRs), while external pathogens donate pathogen-associated molecular patterns (PAMPs) such as lipopolysaccharide (LPS) and injured tissues, or cells release damage-associated molecular patterns (DAMP), such as ATP [4,5]. The interaction between PAMPs/DAMP from pathogens or damaged tissues and TLRs triggers an intracellular signaling event initially managed by myeloid differentiation factor 88 (MyD88) and TIR-domain-containing adaptor-inducing interferon-β (TRIF) [6]. These events lead to the activation of various transcription factors, such as nuclear factor-κB (NF-κB), interferon regulatory factor 3 (IRF3), signal transducer and activator of transcription 3 (STAT-3), and activator protein-1 (AP-1) [7,8,9,10]. By transcriptional activation of these factors, pro-inflammatory genes are newly expressed to stimulate and recruit other immune cells [11,12,13].

Of these transcription factors, AP-1 is a major component in the modulation of inflammatory responses, although it also regulates various cellular responses, such as cell differentiation, cell cycle progression, and apoptosis [14,15]. The activation of AP-1 to express target genes requires dimerization of Fra, c-Fos, and c-Jun families and movement into the nucleus from the cytoplasmic compartment, which is a critical step in the functional role of AP-1 [16,17,18]. Transcriptional activation of nuclear AP-1 in inflammatory responses leads to an increase in the mRNA expression of pro-inflammatory genes, such as cyclooxygenase-2 (COX-2), tumor necrosis factor (TNF)-α, interleukin (IL)-6, and chemokine (C-C motif) ligand 12 (CCL-12) [19]. The translocation step is mediated by their upstream kinases, mitogen-activated protein kinases (MAPKs), such as p38, extracellular signal-regulated kinase (ERK), and c-Jun N-terminal kinase (JNK), a family of serine/threonine protein kinases [20]. Activation of the MAPKs also requires phosphorylation of these proteins [10,21]. Activation of AP-1 allows the transcription of inflammation-related enzymes, including matrix metallopeptidases (MMPs) and cyclooxygenase 2 (COX-2) [22,23]. Phosphorylation of these enzymes is carried out by MAPK kinases (MAPKKs), including MAPK/ERK kinase (MEK1/2) and MAPK kinases (MKK3, 4, 6, and 7). Currently, activation of MAPKKs is mediated by MAPKK kinases (MAP3Ks), notably including transforming growth factor-β-activated kinase 1 (TAK-1), mixed lineage kinase 3 (MLK3), and apoptosis signal-regulating kinase 1 (ASK1) [24,25]. However, their roles in the inflammatory responses of macrophages have not yet been fully understood.

MLK3 (93 kDa) is a serine/threonine/tyrosine kinase with an N-terminal Src-homology-3 (SH3) domain, leucine zipper regions, a Cdc42/Rac-interactive binding motif, and a large C-terminal tail rich in serine, threonine, and proline residues [26,27]. Autoinhibition through the SH3 domain, leucine zipper-mediated dimerization, and transphosphorylation within the catalytic domain are considered regulatory modes of this enzyme [28]. Critically, MLK3 in breast, ovarian, liver, and pancreatic tumors has been reported to activate p38 and JNK, which are involved in the regulation of apoptosis, proliferation, differentiation, migration, invasion, and survival of tumor cells [29,30,31]. Unlike studies of cancer cells, the role of MLK in innate immunity and macrophage-mediated inflammatory responses has not yet been fully elucidated.

Although inflammatory responses are necessary for defensive mechanisms, excessive acute and sustained levels of inflammation also induce serious diseases, such as cancer, atherosclerosis, diabetes, and neuronal diseases [32,33,34,35]. Therefore, developing anti-inflammatory drugs could be a good strategy for preventing such serious disorders. In addition, finding novel target molecules involved in inflammatory signaling cascades will be a good approach to developing novel classes of anti-inflammatory drugs. In this study, we aimed to evaluate the inflammation regulatory role of MLK3 in HEK293 cells and RAW264.7 cells under overexpression and pharmacological inhibition strategies to determine whether this protein could serve as another inflammation-regulatory molecule.

## 2. Results

### 2.1. MLK3 Can Induce Expression of Inflammatory Genes via Activation of AP-1

In order to test whether MLK3 can activate transcription factors found in inflammatory responses, we first employed luciferase reporter gene assays [11,36] performed with DNA constructs such as AP-1-Luc, NF-κB-Luc, STAT3-Luc, IRF-3-Luc, or IFN-γ-promoter-Luc (with GATA/T-bet/NF-AT binding sites [37]). As Figure 1 shows, highly increased levels of MLK3 like Figure 1a strongly enhanced the luciferase activity mediated by AP-1 but not NF-κB, STAT3, IRF-3, and GATA/T-bet/NF-AT (Figure 1b–f). In addition, overexpression of MLK3 significantly stimulated the mRNA expression of COX-2 (23.3 ± 0.8 folds), IL-6 (3.7 ± 0.2 folds), and TNF-α (2289 ± 490 folds), implying that MLK3 can induce pro-inflammatory gene expression via AP-1 activation. However, overexpression of two adaptor molecules found in macrophages, MyD88 and TRIF, did not trigger the expression of MLK3 or HPK1.

### 2.2. MLK3 Can Induce Phosphorylation of MAPKK, MAPK, and AP-1 Subunits

To check whether MLK3 can activate AP-1 and its upstream signaling cascades at the protein level, the phosphorylated forms of MLK3, AP-1 subunits (c-Jun, c-Fos, and FRA1), MAPK (p38, JNK, and ERK), and MAPKK (MKK3, 4, 6, 7, and MEK1/2) were detected by immunoblotting analysis. As Figure 2a shows, overexpressed MLK3 increased the phosphorylation of this protein, MLK3. As expected, phospho-forms of c-Jun, c-Fos, and FRA1 were also enhanced by overexpression of MLK3 (Figure 2b). In agreement with this result, upstream signaling events for the upregulation of p-c-Jun, p-c-Fos, and p-FRA1 were also remarkably increased. Thus, ERK, p38, and JNK were strongly phosphorylated under MLK3 overexpression conditions (Figure 2c). The phosphorylation levels of upstream enzymes to phosphorylate these proteins were also determined. As Figure 2d,e depict, MEK1/2, MKK3/6, and MKK4/7 were found to be phosphorylated, while the phosphorylation level of endogenous TAK1 was not increased. These results seem to indicate that MLK3 can activate MAPKK and MAPK by phosphorylation for the activation of AP-1 subunits.

### 2.3. MLK3 Can Be Activated in RAW264.7 Cells Stimulated by LPS, Pam3CSK, and Poly(I:C)

Next, whether MLK3 can also be activated in macrophages stimulated by inflammation inducers, such as LPS, Pam3CSK, and poly(I:C), was examined by immunoblotting analysis. Indeed, under treatment with LPS, RAW264.7 cells were found to be fully activated according to the mRNA levels of COX-2, IL-6, TNF-α, and IL-1β (Figure 3a–d). Interestingly, the mRNA level of MLK3 was not altered in LPS-treated RAW264.7 cells even at 1 to 6 h (Figure 3e), which showed strong expression levels of inflammatory genes (Figure 3a–e), while the phosphorylation level of MLK3 was strikingly enhanced at 2 to 5 min during exposure to LPS, Pam3CSK, and poly(I:C) (Figure 3f), implying that MLK3 could be controlled at the protein level.

### 2.4. Inhibition of MLK3 Blocks AP-1-Mediated Inflammatory Response in LPS-Treated RAW264.7 Cells

Finally, whether MLK3 can play a critical role in inflammatory responses was investigated using a specific inhibitor (URMC-099, Figure 4a) to MLK3 in LPS-treated RAW264.7 cells and HEK293 cells transfected with TRIF or MyD88. To do this, the cell viability of URMC-099 was first assessed by MTT assay. As Figure 4a,b show, this compound did not display any cytotoxicity up to 12.5 μM in both RAW264.7 (Figure 4b right panel) and HEK293 cells (Figure 4b left panel). Next, we checked whether this compound can block inflammatory responses in LPS-treated RAW264.7 cells by evaluating the expression levels of inflammatory genes, such as COX-2 and CCL-12. As expected, LPS strongly stimulated the mRNA levels of COX-2 and CCL-12, whereas URMC-099 clearly reduced the mRNA levels, as assessed by RT-PCR (Figure 4c). This inhibitory effect was also confirmed by real-time PCR (Figure 4d, left and right panels). Simultaneously, we also evaluated whether this compound can affect AP-1-mediated signaling events by determining the levels of phospho- and total forms of MLK3 and AP-1 (c-Jun) using immunoblotting analysis. As Figure 4e and f displays, URMC-099 clearly blocked the phosphorylation of MLK3 (Figure 4e). Moreover, this compound was confirmed to suppress the early phosphorylation of AP-1 (c-Jun) at 5 and 15 min (Figure 4f). In addition, the functional activity of AP-1 under overexpression of MyD88 or TRIF was reduced by URMC-099 (5 and 10 μM) (Figure 4g). Finally, to test whether this compound can affect upstream signaling for the AP-1 pathway, the levels of p-p38, p-ERK, and p-JNK were determined by immunoblotting analysis. As Figure 4h shows, URMC-099 strongly suppressed their phosphorylation levels, implying that MLK3 could play a critical role in inflammatory responses mediated by AP-1.

## 3. Discussion

In this study, we found that MLK3 plays a central role in the activation of AP-1 via the simultaneous activation of MAPK and MAPKK in HEK293 and macrophage-like RAW264.7 cells. In agreement, the activation of MLK3 was linked to the AP-1-mediated expression of inflammatory genes. These effects were also confirmed by demonstrating the anti-inflammatory activity of an MLK3 inhibitor, URMC-099, in LPS-treated RAW264.7 cells, implying that MLK3 is functionally active in TLR4-mediated inflammatory responses.

Interestingly, the mRNA level of MLK3 did not change with the overexpression of MyD88 and TRIF, implying that MLK3 expression might be differentially controlled in macrophages. Since HPK1 (hematopoietic progenitor kinase 1) and mitogen-activated protein kinase kinase kinase kinase 1 (MAP4K1) to activate JNK [38] also showed no alteration in its expression level under the same conditions, expression control to maintain their level and activity might be managed by inflammation independently. However, treatment with PAMPs, LPS, poly(I:C), and pam3CSK enhanced MLK3 phosphorylation (Figure 3f). This result seems to imply that the phosphorylation and dephosphorylation of MLK3 could be major regulatory mechanisms in inflammatory responses. Thus, it is considered that bacterial and viral infections can induce the activation of MLK3 via balancing phosphorylation and dephosphorylation of this protein. Indeed, it has been reported that autoinhibition and autophosphorylation are regulatory mechanisms of MLK3 [28]. Recently, ERK has been considered an upstream MLK3 phosphorylation-inducing enzyme in colon cancer cells [39]. However, which protein can trigger its phosphorylation in inflammatory cells is not currently clear, and further study is required to fully understand the regulatory loop of this protein through cooperation with other signaling molecules. In addition to MLK3, since other MAP3K and their upstream enzymes MAP4K involved in AP-1 activation have not yet been fully elucidated in macrophage-mediated inflammatory responses, additional experiments are required to understand these pathways.

Although we found that MLK3 plays an important role in the modulation of inflammatory signaling, the next step will be to confirm whether the inhibition of MLK3 is linked to anti-inflammatory outcomes. To suppress MLK3, researchers have used broad spectrum inhibitors, such as URMC-099, an orally bioavailable brain penetrant mixed lineage kinase (MLK) inhibitor with an IC_50_ value of 14 nM [40]. Currently, few papers have reported on the in vitro and in vivo anti-inflammatory activities of URMC-099. For example, URMC-099 was revealed to protect orthopedic surgery-triggered neuroinflammation (microgliosis) and memory impairment without affecting fracture healing in a perioperative neurocognitive disorder mouse model [41]. A protective effect of hippocampal synapses by this compound in experimental autoimmune encephalomyelitis (EAE)-induced mice was also observed [42]. In four-month-old APP/PS1 mice with Alzheimer’s disease (AD), it was found that URMC-099 can restore synaptic integrity and hippocampal neurogenesis via facilitating Aβ clearance in the brain [43], implying multifaceted immune modulatory and neuroprotective roles of URMC-099. Moreover, URMC-099 was also found to reduce the inflammatory response of microglial cells and enhance the phagolysosomal degradation of Aβ via increased scavenger receptors. Finally, it was also reported that URMC099 suppresses the migration of breast cancer cells but not cell growth in vitro or tumor formation in a mouse breast cancer xenograft model [44]. Similarly, this compound suppressed the recruitment of neutrophils into the peritoneal cavity stimulated by fMLP, demonstrating the role of MLK3 in cell migration [45]. Fully considered, these results suggest that URMC-099 is effective in brain diseases by modulating microglial cells. However, based on our results, this compound could also show anti-inflammatory activity against AP-1-mediated inflammatory diseases. Since the AP-1 pathway is critical in various acute and chronic diseases, such as septic shock, arthritis, gastritis, and colitis [6,46,47,48], URMC-099 may be an effective drug to treat these diseases. We plan to further test the potential activity of URMC-099 in the following projects.

## 4. Materials and Methods

### 4.1. Materials

URMC-099 was obtained from Sigma Aldrich Co. (St. Louis, MO, USA). RAW264.7 and HEK293T cells were purchased from the American Type Culture Collection (Rockville, MD, USA). Fetal bovine serum (FBS), Dulbecco’s Modified Eagle Medium (DMEM), and phosphate-buffered saline (PBS) were purchased from Gibco (Grand Island, NY, USA). 3-(4-5-Dimethylthiazol-2-yl)-2,5-diphenyltetrazolium bromide (MTT) was purchased from Amresco (Brisbane, Australia). Lipopolysaccharide (LPS), Poly (I:C), Pam3CSK4, polyethylenimine (PEI), and dimethyal sulfoxide (DMSO) were purchased from Sigma Chemaical Co. (St. Louis, MO, USA). TRIzol and PCR premix were purchased from Bio-D Inc. (Seoul, Korea). cDNA synthesis kits were purchased from Thermo Fisher Scientific (Waltham, MA, USA). Forward and reverse primers for PCR (polymerase chain reaction) were synthesized by Macrogen, Inc. (Seoul, Korea). All antibodies related to the phosphorylated or total forms of the target protein were purchased from Cell Signaling Technology (Beverly, MA, USA) and Santa Cruz Biotechnology, Inc. (Santa Cruz, CA, USA).

### 4.2. Cell Cultures

Mouse-derived RAW264.7 macrophage cell line and human-derived HEK293T embryonic kidney cells were cultured in RPMI 1640 with 10% FBS and 1% antibiotics (streptomycin and penicillin). HEK293T cells were cultured in DMEM with 5% FBS and 1% antibiotic. These cell lines were incubated in 5% CO_2_ at 37 °C.

### 4.3. Cell Viability Tests

RAW264.7 and HEK293T cells were seeded at 2 × 10^5^ cells per well in 96-well plates and incubated overnight for enough confluency. Then, different concentrations of URMC-099 (0–12.5 μM) were added to them, and then they were incubated for 24 h. Incubated cells were treated with 10 μL/well of MTT solution, and after 3 h, they were treated with 100 μL of MTT Stopping Solution. Using the conventional MTT assay for measuring cell viability [49,50], the absorbance at 570 nm was measured using a reader (BioTek Instruments, Winooski, VT, USA).

### 4.4. mRNA Analysis by Semi-Quantitative RT-PCR and Quantitative Real-Time PCR

RAW264.7 cells were treated with LPS (1 μg/mL) in the presence or absence of URMC-099 (0–10 μM) for 6 h, and HEK293 cells were transfected with Myc-MLK3 (1 μg/mL) in the presence or absence of co-transfected MyD88 or TRIF for 24 h. RNA was then prepared from these cells using TRI reagent, as reported previously [51]. A cDNA synthesis kit was used to synthesize the complementary DNA. RT-PCR (the mRNA expression levels of COX-2, CCL12, and GAPDH in RAW cells) and real-time PCR (the mRNA expression levels of COX-2, TNF-α, IL-1β, IL-6, CCL-12, MLK3, and HPK1) were conducted using specific reverse and forward primers, as reported previously [52,53]. Primers for RT-PCR and real-time PCR are listed in Table 1.

### 4.5. Immunoblotting Analysis

RAW264.7 and HEK293T cells were seeded at a density of 1 × 10^6^ cells/mL and 3 × 10^5^ cells/mL, respectively. The HEK293T cells were further treated with MYC and p-CMV-MYC-MLK3 for 24 h. URMC-099-treated RAW264.7 cells were also incubated with LPS, Pam3CSK, or poly(I:C) for the indicated times. The cells were washed in 1 mL of cold PBS and collected with a cell lysis buffer. Protein preparation and whole cell lysates were performed from these cells, as described previously [49,54]. The cell lysates were centrifuged, and the subsequent supernatant was used for Western blotting analysis. Specific antibodies were used to detect the total and phosphorylated forms of c-Jun, c-Fos, FRA-1, JNK, p38, ERK, MKK3/6, MKK4/7, TAK1, MLK3, and β-actin, which were visualized with chemiluminescence reagents [55].

### 4.6. Luciferase Reporter Gene Assay

HEK293T cells were prepared with a density of 3 × 10^5^ cells/mL and divided into 24-well plates. The HEK293T cells were transfected with 1 μg of plasmids containing MLK3, adaptor molecule (MyD88 or TRIF), AP-1-Luc, NF-κB-Luc, IRF3-Luc, STAT-3-Luc, or IFN-γ-Prom-Luc (with GATA/T-bet/NF-AT binding sites), and β-galactosidase in the presence or absence of URMC-099, employing polyethylenimine [56,57]. After 24 h of incubation, the cells were harvested. The luciferase assay was performed with a luminometer, and the absorbance of each sample was measured at 475 nm using a Spectramax 250 microplate reader (Molecular Devices, San Jose, CA, USA), as reported previously [58].

### 4.7. Statistical Analysis

At least three independent experiments were repeated to obtain the data and are presented as the mean ± standard deviation for the concise results. In order to judge statistical differences between groups, the Mann–Whitney test, with a *p*-value < 0.05 to be considered statistically significant, was employed.

## 5. Conclusions

In summary, we found that MLK3 can act as a critical molecule in the AP-1-mediated inflammatory response of macrophages, as summarized in Figure 5. Thus, overexpression of MLK3 upregulated AP-1 activity and increased mRNA expression levels of COX-2, IL-6, and TNF-α. Overexpression of MLK3 also enhanced the phosphorylation of AP-1 subunits and upstream components of MAP2K and MAPK (c-Jun, c-Fos, and FRA-1). This activated RAW264.7 cells stimulated by LPS, Pam3CSK, and poly(I:C); phospho-MLK3 levels also increased at early time points. Indeed, suppression of this enzyme by URMC-099 reduced the mRNA expression of COX-2 and CCL-12, phosphorylation of c-Jun, luciferase activity mediated by AP-1, and phosphorylation of MAPK in LPS-treated RAW264.7 cells. Therefore, our results clearly imply that MLK3 acts as a central enzyme regulating AP-1-mediated inflammatory responses in macrophages. Since prolonged and severe inflammation responses are considered the cause of numerous serious diseases, this enzyme can serve as a target molecule for treating AP-1-mediated inflammatory diseases.

## Figures and Tables

**Figure 1 ijms-23-10874-f001:**
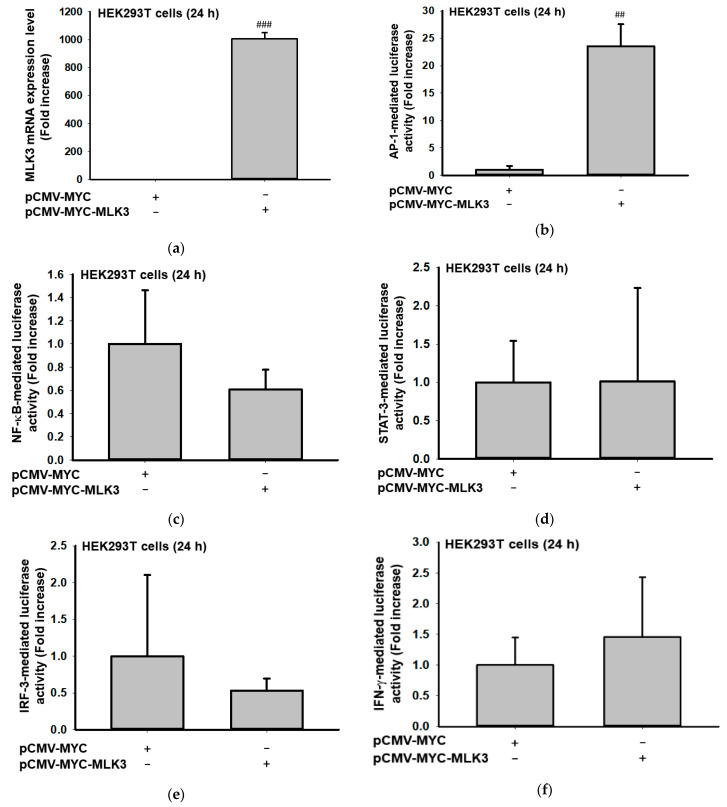
Effect of MLK3 on AP-1 activation. (**a**,**g**,**h**) The mRNA expression levels of MLK3 and HPK1, as well as inflammatory genes (COX-2, IL-1β, IL-6, and TNF-α) were determined by real-time PCR from HEK293 cells transfected with Myc-MLK3 (**a**,**g**) or adapter molecules (MyD88 and TRIF) (**h**). (**b**–**f**) Luciferase activity mediated by AP-1, NF-κB, IRF3, and GATA/T-bet/NF-AT in HEK293 cells transfected with Myc-MLK3 and AP-1-Luc, NF-κB-Luc, STAT3-Luc, IRF-3-Luc, or IFN-γ-promoter-Luc was determined by luminometer. Results (**a**–**f**) are expressed as mean ± SD. # *p* < 0.05, ## *p* < 0.01, and ### *p* < 0.001 compared to the normal group (no treatment). MYC: empty vector (PCMV) with MYC.

**Figure 2 ijms-23-10874-f002:**
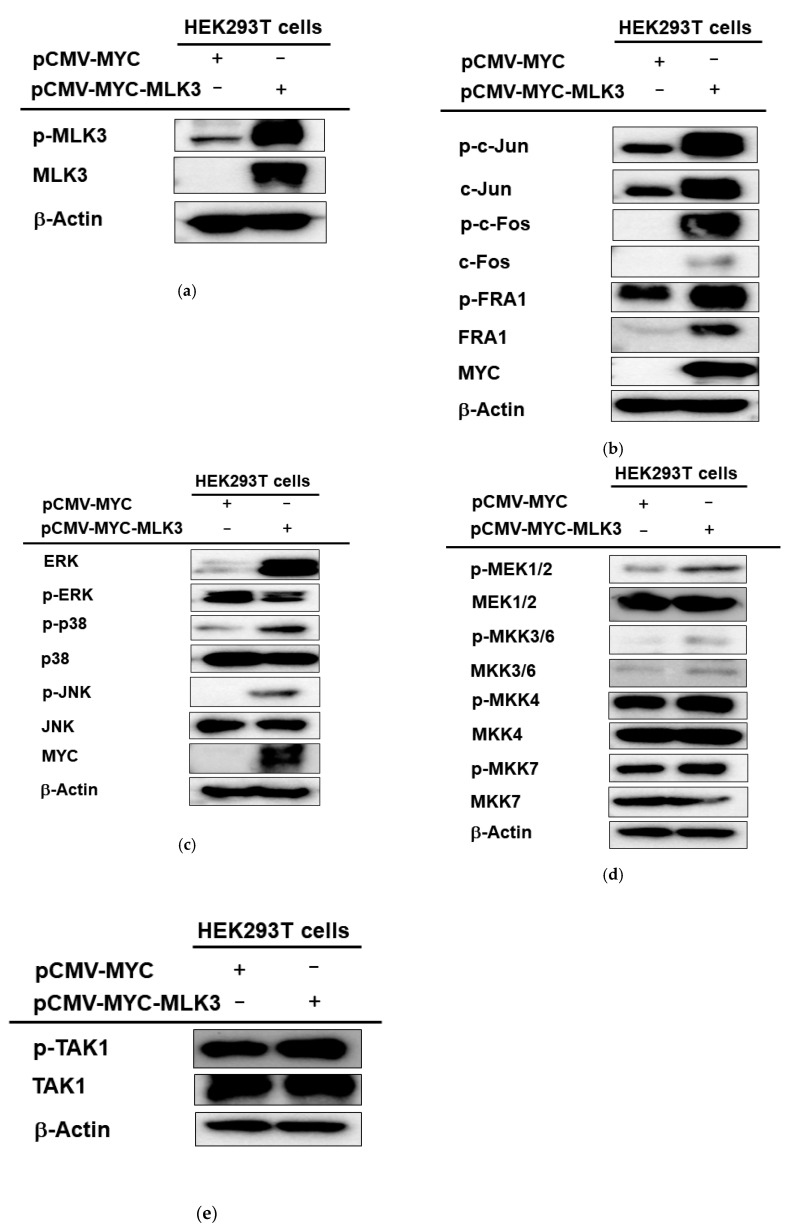
Effect of MLK3 on the phosphorylation of AP-1 subunits, MAPK, and MAPKK. (**a**–**e**) The phospho- and total forms of MAPK, MAPKK, and AP-1 subunits were detected from whole cell lysates of HEK293 cells transfected with Myc-MLK3 (1 μg/mL) for 36 h by immunoblotting analysis.

**Figure 3 ijms-23-10874-f003:**
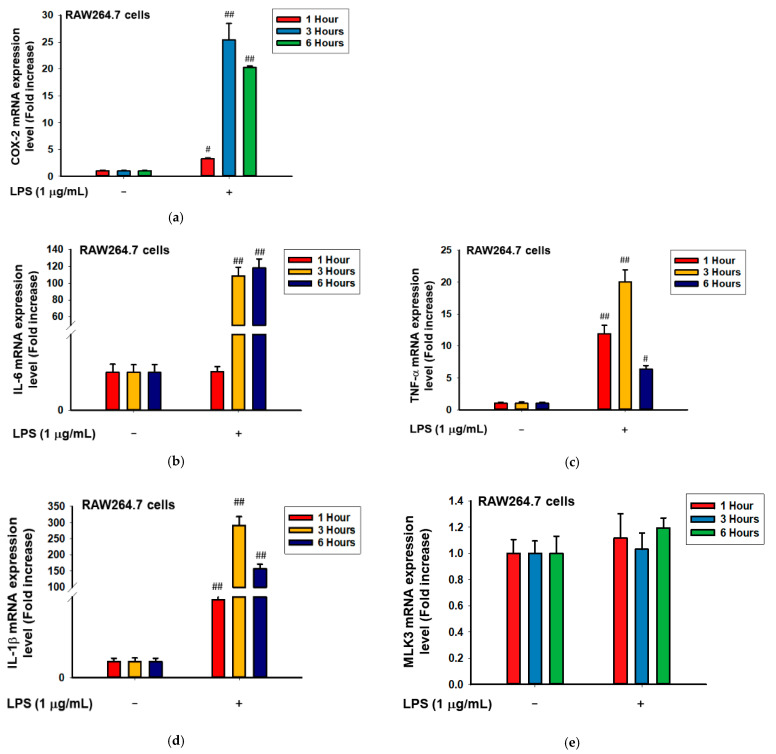
MLK3 is phosphorylated in activated RAW264.7 cells during exposure to LPS, Pam3CSK, and poly(I:C). (**a**–**e**) The mRNA expression levels of MLK3 and inflammatory genes (COX-2, IL-1β, IL-6, and TNF-α) were determined by real-time PCR from RAW264.7 cells stimulated with LPS. (**f**). The phospho- and total forms of MLK3 were detected in whole cell lysates of RAW264.7 cells stimulated with LPS, Pam3CSK, and poly(I:C) by immunoblotting analysis. Results (**a**–**e**) are expressed as mean ± SD. # *p* < 0.05 and ## *p* < 0.01 compared to the normal group (no treatment).

**Figure 4 ijms-23-10874-f004:**
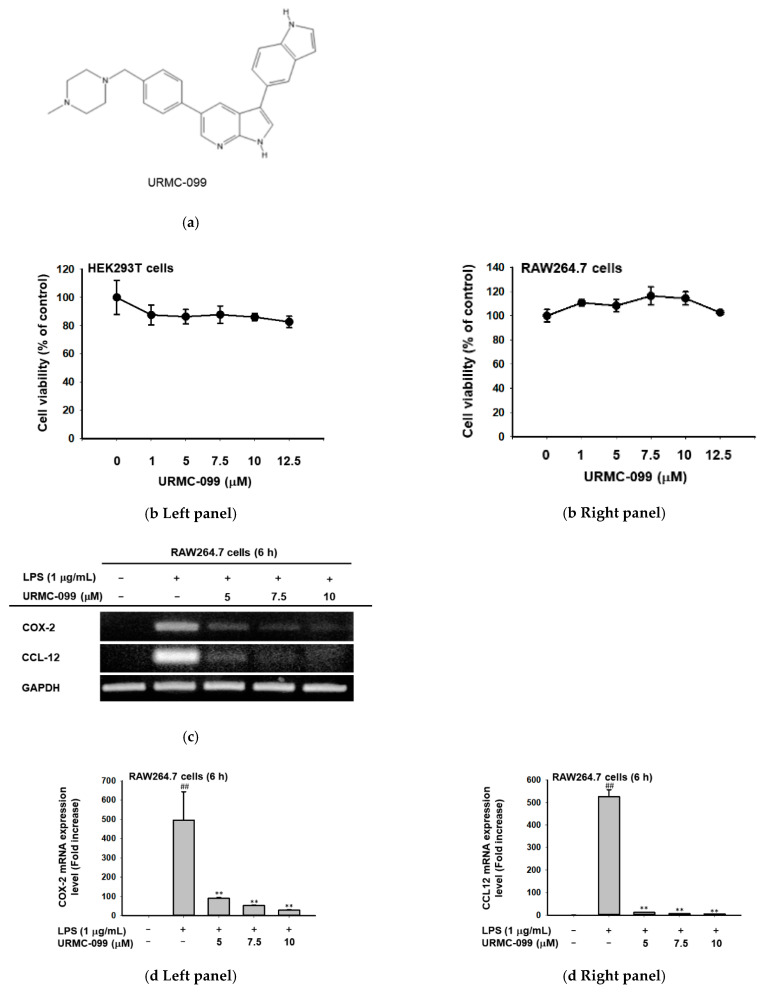
Inhibition of MLK3 reduces inflammatory responses mediated by AP-1 in LPS-activated RAW264.7 cells. (**a**) Chemical structure of URMC-099. (**b**) Viability of RAW264.7 and HEK292 cells under URMC-099 treatment conditions was evaluated by MTT assay. (**c**,**d**) The mRNA expression levels of COX-2 and CCL-12 were determined by RT-PCR and real-time PCR from RAW264.7 cells stimulated with LPS. (**e**,**f**,**h**). The phospho- and total forms of MLK3, c-Jun, JNK, p38, and ERK were detected from whole cell lysates of RAW264.7 cells stimulated with LPS by immunoblotting analysis. (**g**) Luciferase activity mediated by AP-1 in HEK293 cells transfected with AP-1-Luc as well as TRIF or MyD88 in the presence or absence of URMC-099 was determined by luminometer. Results (**b**,**d**,**g**) are expressed as mean ± SD. # *p* < 0.05 and ## *p* < 0.01 compared to the normal group (no treatment) and * *p* < 0.05 and ** *p* < 0.01 compared to the control group (LPS, MyD88, or TRIF alone).

**Figure 5 ijms-23-10874-f005:**
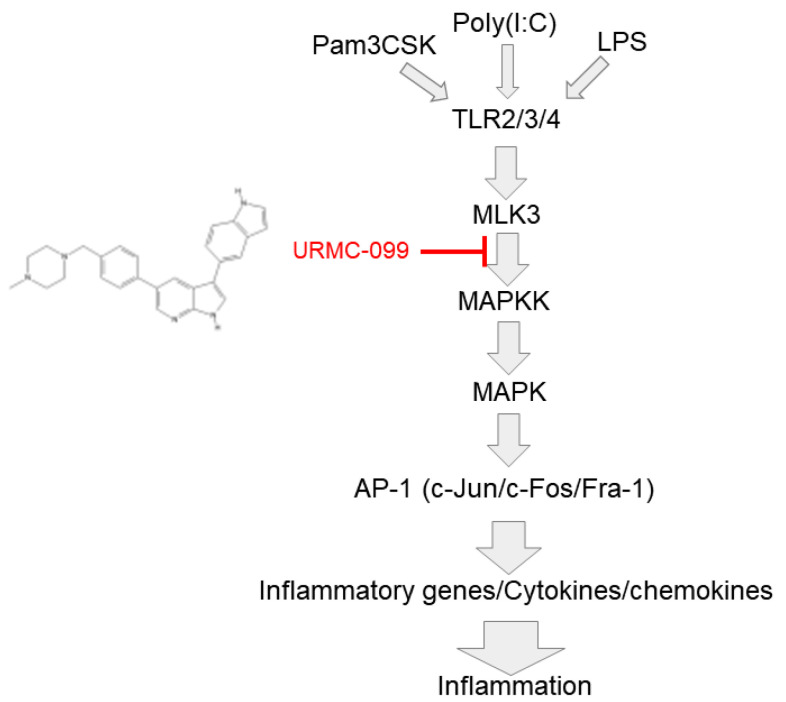
Schematic summary of inflammation-regulatory role of MLK3.

**Table 1 ijms-23-10874-t001:** Primer sequences for the analysis of mRNA prepared for RT-PCR and real-time PCR.

Name	Direction	Sequence (5′ to 3′)
Primer Sequences used in RT-real-time PCR		
*COX-2*	Forward	TTGGAGGCGAAGTGGGTTTT
	Reverse	TGGCTGTTTTGGTAGGCTGT
*TNF-α*	Forward	TGCCTATGTCTCAGCCTCTT
	Reverse	GAGGCCATTTGGGAACTTCT
*IL-1β*	Forward	GTGAAATGCCACCTTTTGACAGTG
	Reverse	CCTGCCTGAAGCTCTTGTTG
*IL-6*	Forward	AGCCAGAGTCCTTCAGAGAGAT
	Reverse	AGGAGAGCATTGGAAATTGGGG
*CCL-12*	Forward	GCCTCCTGCTCATAGCTACC
	Reverse	CTTCCGGACGTGAATCTTCT
*MLK3*	Forward	GTCGACAATGGAGCCCTTGAAGAGCCTC
	Reverse	CGGCCGTCAAGGCCCCGCTTCCG
*HPK1*	Forward	CTGCTGGAACGGAAAGAGAC
	Reverse	CGGACAAGCAGGAATTTGTT
*GAPDH*	Forward	TGTGAACGGATTTGGCCGTA
	Reverse	ACTGTGCCGTTGAATTTGCC
Primer Sequences used in RT-PCR		
*COX 2*	Forward	TCACGTGGAGTCCGCTTTAC
	Reverse	CTTCGCAGGAAGGGGATGTT
*CCL-12*	Forward	GCCTCCTGCTCATAGCTACC
	Reverse	CTTCCGGACGTGAATCTTCT
*GAPDH*	Forward	CACTCACGGCAAATTCAACGGCA
	Reverse	GACTCCACGACATACTCAGCAC

## Data Availability

The data used to support the findings of this study are available from the corresponding author upon request.

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
