# Peer review of "MLK3 Regulates Inflammatory Response via Activation of AP-1 Pathway in HEK293 and RAW264.7 Cells"

_ijms, 2022, doi:10.3390/ijms231810874_

Round 1

Reviewer 1 Report

The manuscript of Anh Thu Ha and colleagues describes the regulation of inflammatory response in different cell lines and it is focused on the role of mixed lineage kinase 3 (MLK3) in regulating the enzyme AP-1. The manuscript is interesting and well-written.

Comments:

-          Figure 1a, the relative graph is incomplete. The column on the right side is not completely visible.

-          In Figure 1h, right panel the last column is not completely visible.

-          In figure 2a, MLK3 seems absent in HEK293T cells while a small band is present in the blot for p-MLK3. May the authors give an explanation for this? Do you have a higher exposure of the blot? I would also like to ask you whether you have a lower exposure for p-FRA1 or a higher one for FRA1.

-          In figure 2e, do you have a lower exposure for p-MLK3 bands?

-          Figure 3f – abundance of the respective total forms is missing.

-          In the supplementary data the names of the samples and the molecular weights on the left of the blots should be reported.

Reviewer 2 Report

Please see attached

Round 2

Reviewer 1 Report

I consider the manuscript suitable for publication.

Reviewer 2 Report

All the comments have been addressed